# ‘Teratoid’ Hepatoblastoma: An Intriguing Variant of Mixed Epithelial-Mesenchymal Hepatoblastoma

**DOI:** 10.3390/children9040565

**Published:** 2022-04-15

**Authors:** Consolato M. Sergi, Marta Rojas-Vasquez, Michelle Noga, Bryan Dicken

**Affiliations:** 1Anatomic Pathology Division, Children’s Hospital of Eastern Ontario, Ottawa, ON K1H 8L1, Canada; 2Department of Lab. Medicine and Pathology, Stollery Children’s Hospital, University of Alberta, Edmonton, AB T6G 2B7, Canada; 3Department of Pediatric Hematology-Oncology, Stollery Children’s Hospital, University of Alberta, Edmonton, AB T6G 2B7, Canada; marta.rojas-vasquez@albertahealthservices.ca; 4Department of Pediatric Radiology, Stollery Children’s Hospital, University of Alberta, Edmonton, AB T6G 2B7, Canada; mnoga@ualberta.ca; 5Department of Surgery, Stollery Children’s Hospital, University of Alberta, Edmonton, AB T6G 2B7, Canada; bdicken@ualberta.ca

**Keywords:** liver, cancer, radiology, management

## Abstract

Liver neoplasms are quite rare in childhood. They often involve 6.7 cases per 10 million children aged 18 years or younger. Hepatoblastoma (HB) is the most frequent tumor, but this neoplasm’s rarity points essentially to the difficulty of performing biologic studies and large-scale therapeutic trials. On the pathological ground, HB is separated into an entirely epithelial neoplasm or a mixed neoplasm with epithelial and mesenchymal components. This last category has been further subdivided into harboring teratoid features or not. The ‘teratoid’ HB includes a mixture of components with heterologous origin. The heterologous components include neuroectoderm, endoderm, or melanin-holding cells with or without mesenchymal components. The most important criterium for the teratoid component is neuroepithelium, melanin, and, more recently, a yolk-sac-like component and neuroendocrine components. The mesenchymal components include muscle, osteoid, and cartilage, which are most often observed mainly in ‘teratoid’ neoplasms. The teratoid component or mesenchymal components are diagnosed with biopsies. They appear more prominent after chemotherapy due to the response and shrinkage of epithelial elements and non- or low-responsive components of mixed HB. This review focuses on the clinical, radiological, and pathological findings of HB with teratoid features.

## 1. Introduction

The challenges of a pediatric oncologist facing a liver neoplasm are due to the extreme difficulty of performing biologic studies and large-scale curative trials, despite there being a nationwide consortium of biological researchers and multiple clinical investigators. Hepatoblastoma (HB) is the most customary liver neoplasm in patients aged 18 years or younger, with a rate of one per approximately 1–1.5 million children [1,2]. The most common components of this tumor are epithelial and mesenchymal (Figure 1). In Figure 1, it shows the most common classification of HB used in the United States, Canada, Europe, Asia, Australasia, and Africa.

The increased interest in HB arises from the numerous recent studies on experimental animals. These studies connected HB to pesticides and herbicides and the introduction of new therapeutic strategies for drug repurposing. The current and intense investigation of *1-tert-butoxypropan-2-ol* as a constituent of saleable cleaner formulations, *pyridine* as a flavoring agent, and *β-myrcene* as a fragrance and flavoring agent by the International Association for Research on Cancer (IARC), an international agency of the World Health Organization (WHO), has drawn impressive and unprecedented attention of the scientific community for the increased incidence of HB in experimental animals exposed to these molecular compounds [3,4,5,6]. Since some of these agents are widely spread in our societies as vaping agents, it may be necessary to consider setting up appropriate epidemiological studies in the future. The concept of drug repurposing is linked to the feasibility of using a conventionally used drug for other cancers [7]. In the previous four decades, the application of genomics to enlighten drug advancement pipelines has triggered both enthusiasm and skepticism. Initially, early efforts have profitably discovered some innovative drug targets. Still, the whole clinical usefulness, in terms of efficiency and effectiveness of the developed drugs, has lingered relatively mediocre, and this evolution may have its roots, in our opinion, in the heterogeneous signaling and etio-pathogenesis of cancer.

Nevertheless, most recent high-tech and analytical developments in genomics, proteomics, and metabolomics, have prompted the rapid identification and interpretation of several genetic variations underlying even a single affected individual’s disease, thereby paving the road for precision medicine perspectives in the 21st century [8]. Genomic-wide association studies (GWAS) have paved the road to identifying new genetic determinants from a drug discovery standpoint. It supplied valuable expertise and knowledge about the genetic design of illness and the prospective indicators of inherent causative mechanisms. It stimulated the pledge of uncovering better candidate targets for rare and less rare diseases [9,10,11]. Stimulated, both by the experimental investigations and new platforms and venues in drug discovery and research, as well as reviewing and paneling for the Children’s Oncology Group (COG) and the International Society of Pediatric Oncology (SIOP), we propose, here, a clinic-radiologic and pathological review of a very rare variant of HB, i.e., the ‘teratoid’ HB.

### 1.1. Roots of the ‘Teratoid’ Concept

The concept and, specifically, the term ‘teratoid’ hepatoblastoma or hepatoblastoma with ‘teratoid’ features was first advanced by Manivel et al. more than three decades ago [12], even if an analogous lesion had been previously described in 1965 [13]. The identification of deviating tissue, such as osteoid or even frank bone, is a well-recognized occurrence in the mesenchymal HB, and these divergencies in the differentiation are more often detected in the post-chemotherapy liver specimens (hepatectomy). The presence of a combination of heterologous elements, including endoderm, neuroectoderm, neural crest elements (e.g., ganglion cells and melanocytes), or melanin-containing cells, correctly characterizes the mixed epithelial designation and mesenchymal tumor as ‘teratoid’ HB [1,14,15]. Thus, the term ‘teratoid’ should be reserved for mixed HBs harboring tissues originating from all three germ layers. These elements include cartilage, muscle (skeletal), keratinizing multistratified squamous epithelium, intestinal epithelium, bronchial epithelium, and melanin pigment [16]. In four-fifths of mixed tumors, it has been detected that the mesenchymal component is represented by “fibrous” tissue of immature or primitive type, cartilage, and/or osteoid. At the same time, the remaining 20% are mixed HB with teratoid features that may include additional tissue phenotypes, such as intestinal-type glandular elements, melanin pigment, classic skeletal muscle, neural tissue, or melanin pigment, which can be enhanced using DOPA (3,4-dihydroxyphenylalanine) oxidase, ferrous iron, Fontana-Masson, and Fontana-Masson picrosirius methods. The most important criteria for the teratoid component remain the neuroepithelium, melanin, and, more recently, the yolk-sac-like component and neuroendocrine components. The mesenchymal components include muscle, osteoid, and cartilage, which are most often observed mainly in ‘teratoid’ neoplasms. The teratoid component or mesenchymal components are diagnosed with biopsies. They appear more prominent after chemotherapy due to the response and shrinkage of epithelial elements and non- or low-responsive components of mixed HB. The response to chemotherapy in these neoplasms of the liver is quite variable, with some tumors displaying complete resolution and some persistence post-therapy. It is important to emphasize that it is the epithelial component that determines the aggressiveness of the HB, with the teratoid component only rarely being demonstrated in metastases. 

### 1.2. Clinical-Radiological Features

Primary malignant neoplasms of the liver are sporadic neoplasms, contributing to about 1% of all childhood malignancies. The most common types are HBs, hepatocellular carcinomas (HCCs), and sarcomas, of which some are most often seen before the age of three and some after this age. Some are associated with specific genetic variations, such as the fibrolamellar carcinoma—also called the fibrolamellar variant of HCC [17,18]. In 90% of HBs, this tumor occurs in the first and second infancy (first five years of life), and 5% of these neoplasms are congenital. There is a perceptible trend in the last three decades (1990’s–2020) of increased incidence of HB in infants born prematurely or with a very low birth weight [19,20,21]. Our environmental considerations may need further studies in this direction. There are several genetic constellations and syndromes associated with HB, including familial adenomatous polyposis (FAP), Beckwith–Wiedemann syndrome (BWS), trisomy 13, 18, and 21, and partial trisomy 9p [19,20,21,22,23,24,25,26,27]. From the clinical standpoint, four-fifths of patients present with a single mass, while one-fifth present with multiple nodules. Although rare, the metastatic spreading ability of HB has been described in both humans and animals [28,29,30,31,32], and human spreading intumescences occur most frequently in the respiratory tract (lungs) and, rarely, in the brain, choroid, iris, and skin [33]. Radiologically, HB of mixed epithelial and mesenchymal types is usually an inhomogeneous mass detected by magnetic resonance tomography (MRT) (Figure 2). Most often, the tumor presents with two main components, and the coronal view shows that the neoplasm may be located predominantly outside the liver. Equally, the transversal view repeatedly demonstrated that one part has a cystic appearance, whereas another appears more solid. It is advantageous to perform the T1 sequence carried out after a contrast application because it shows an inhomogeneous contrast enhancement with one section of the tumor enhancing more than other sections. 

### 1.3. Pathological Features

On the pathological ground, HB is separated into several histological phenotypes. They are based on patterns of differentiation. They include pure fetal epithelial, combined fetal and embryonal epithelial, macrotrabecular, mixed epithelial and mesenchymal, and mixed epithelial and mesenchymal with teratoid features [1]. The ‘small cell undifferentiated’ (SCUD) variant is no longer shown to be of significance regarding the prognosis of HB. Most of the pure SCUD neoplasms are malignant rhabdoid tumors. The resemblance of the tumor to the developing liver helps in identifying the fetal component, while more primitive areas suggest the diagnosis of an embryonal pattern. The identification of neural, or melanocytic cells, helps restrict the diagnosis of HB with teratoid features (Figure 3 and Figure 4). The ‘teratoid’ HB includes a mixture of heterologous features, such as the endoderm, neuroectoderm, or even melanin-encompassing cells with or without mesenchymal components, such as fibroblastic stroma, muscle, cartilage, or osteoid. These elements are most often notably observed in neoplasms following chemotherapy. The ‘teratoid’ HB or HB with teratoid features is a rare histologic subtype of the mixed epithelial-mesenchymal category of HB, accounting for 4% to 10% of all HBs. The presence of divergent differentiation, including neural and melanocytic, in addition to cartilaginous, osseous, skeletal muscle, and neural elements, has suggested that this tumor might originate from stem cells. The teratoid HB is interesting because of the induction of stem cells in the histogenesis of this kind of pediatric liver neoplasm [34]. 

Some HB may also harbor “cholangioblastic” features, which may entail uni- or bipotential neoplastic cells of the progenitor level, capable of shifting into either of two lines along with the progression of a certain tumor recalling the development of the ductal plate [35,36,37,38,39,40]. Cholangioblastic HB is a variant of epithelial HB and should not be considered part of the ‘teratoid’ concept. Increased recognition of the cholangioblastic component of an epithelial HB is crucial to address beyond using molecular studies and to anticipate our knowledge of the pathogenesis of HB. 

The ‘teratoid’ elements might originate from either multipotent or, probably, less-committed cells (stem cells) [41], and chemotherapeutic approaches could provoke differentiation of the less differentiated cells, as often observed in nephroblastoma or Wilms’ tumor [42]. Immunohistostaining inconsistencies may be observed in HB, and this aspect is triggered by fluctuating epithelial, mesenchymal, and teratoid cell populations, varying intracytoplasmic constituents, such as glycogen and lipid, or cellular “anaplasia” [14,15,16]. Substantially, α-fetoprotein (AFP) is traditionally the most consistent marker found in HBs, which should also be accompanied by the expression of glypican 3, a serum marker for HB [43,44]. The Ruck et al. “small epithelial cells” with a distinguishing phenotype between the hepatic and biliary cells, carried out by immunohistochemistry and electron microscopy, are probably crucial for understanding the evolution from ontogenesis to oncogenesis [45]. These cells, which are denominated ‘oval cells’, express cytokeratin 7 (CK-7), albumin, and oval cells 6 (OV-6). The concept of oval cells has traditionally been applied to mouse models of liver cell regeneration, but the equivalent of the oval cells in humans remains poorly understood, or even unknown. These cells are predominantly found in neo-proliferative processes with anaplastic features, declining in embryonal or fetal differentiated HB. The existence of OV-6 positive cells in HB would point toward the presence of “oval cell progenitors” as the main elements of the ‘teratoid’ HB, but these concepts are heavily debated among liver pathologists and fetal liver experts. In the adult liver, oval cells (“hepatic stem cells”) with a “clonogenic” potential and a dual differentiation potential (either into hepatic or biliary cell lineages) may have been recognized since 1996 [46,47], although there is no uniformity of opinion on this. These stem cells are thought to be bone marrow-derived, though the putative liver stem cell is not yet identified with certainty. Clonogenicity refers to the ability of a determinate cell to progressively clone itself and ultimately grow into a complete colony of cloned cells. Oval cells show a distinctive co-expression of immunophenotypic stem cell markers (e.g., CD34, Thy1, and some others) concurrently with hepatic lineage cell markers (e.g., K-18 or CK-18, albumin, and others) [48,49]. The ‘immature-looking cells,’ suggested by Zimmermann [35], differ from the “true” small cell undifferentiated (SCUD)-HB cells. As indicated above, SCUD is no longer shown to be of significance regarding the prognosis of HB and most of the pure SCUD neoplasms are malignant rhabdoid tumors. These cells, with a potential toward several lines of differentiation, harbor reduced expression of hepatocyte antigen, stronger nuclear/cytoplasmic expression of β-catenin, and a decreased proliferation activity (Ki-67 labeling index) than the other cells of epithelial type. In his review, Zimmermann regarded these cells as one of the epithelial components and ‘pacemaker cells’ of the neoplasm [35]. Probably, the cells that Zimmermann was referring to correspond to the so-called blastemal cells in HB and are capable of bidirectional differentiation. The primitive glandular epithelial component is usually positive for periodic acid–Schiff (PAS) and mucicarmine, suggesting mucin production, and being positive for cytokeratin 19 (K19 or CK19)—an early bile duct lineage differentiation marker of the intermediate cytoskeleton, B-cell lymphoma 2 (BCL-2), which is a well-known anti-apoptotic protein—(cyto-)keratin 20 (K20 or CK20), and β-catenin, and focally positive for K7 (CK-7), but negative for glypican 3, delta-like protein, and claudin 1. Conversely, the embryonal and fetal components of the HB surrounding the primitive glandular epithelium are usually positive for β-catenin, glypican 3, delta-like protein, and claudin 1 but negative for K19 (CK19), BLC2, K20 (CK20), and K7 (CK7). Typically, both the embryonal components and the primitive glandular epithelium are negative for chromogranin A, neuroendocrine, and synaptophysin, as well as other neural markers. An important conundrum remains the vague differentiation of ‘teratoid’ HB from hepatic teratoma, at least for some authors. We consider teratomas rare neoplasms with a rate of 7 cases per million children per year with tissue originating from all three germ layers. In the liver, four well-characterized teratomas have been reported [50,51,52,53]. Moll’s combined HB and teratoma case report should be considered a ‘teratoid’ HB [54]. The distinction of ‘teratoid’ HB from a teratoma needs the presence of an epithelial HB component to formulate that diagnosis. 

### 1.4. Management

Although traditionally challenging, neoadjuvant chemotherapy for HB usually results in a substantial reduction in the tumor size. Archetypal adjustments include neoplastic cell necrosis, a fibro-histiocytic response, “peliosis”-like areas, cytoarchitectural delineation mimicking nonneoplastic hepatocytes and biliary segments, and HCC–like histologic changes [55]. It is important to emphasize that the teratoid HB responds poorly to chemotherapy. Thus, its recognition has prognostic implications. In Buccoliero’s patient, the residual neoplasm was resected fifteen months later, and the surgical specimen contained teratoid features [56]. These teratoid areas have been first documented by Forouhar et al. in 1984 [57], but Kim has often reported in 2001 [34]. Currently, mixed HB remains difficult to prognosticate, despite some authors considering they harbor a relatively good prognosis with a 5-YSR of 70%. However, mixed HB’s prognosis does not approach the 5-YSR of 90–100% with the pure fetal HB, which is low mitotically active. In most cases of mixed HB, the prognosis is determined by the epithelial components and, more often, an embryonal component or pleomorphic epithelial component. Mesenchymal components can be seen in metastatic tumors, but they are usually a small component and intermixed with the epithelial component. The current approach to hepatoblastoma is depicted in Figure 5 and relies essentially on a modified flow-chart of Hiyama [58]. 

Curative surgery is the primary therapy for all pediatric liver tumors, particularly in the lack of metastatic neoplastic disease. Systemic chemotherapy remains advantageous as metastases are diagnosed in about one-fifth of all pediatric patients. Most children suffer from neoplasm recurrence after surgery alone. Clinical data that are provided to the pathologist include: age, α-fetoprotein levels, underlying hepatic disorders, imaging investigational studies (with and without contrast), staging (PRETEXT), maternal and family histories, birth weight (e.g., prematurity, small for gestational age), inherent viral diseases, as well as developmental, genetic, and metabolic disorders. This information is crucial to address the diagnosis adequately. Primary resection of the neoplasm is proposed only for the PRETEXT I tumor. Still, a diagnostic biopsy is proposed for all patients with the intent to have a histological diagnosis classification and start biological and potentially translational studies of the COG and SIOP. The recognition of well-differentiated fetal HB is crucial to avoid redundant chemotherapy to deal with these patients. Fresh liver specimens should be sent to the surgical pathologist (liver pathologist, pediatric pathologist) whenever possible. It has been recommended that a fine-needle aspiration (FNA) should be avoided for diagnosis. It is done in this way because the material obtained by FNA is usually insufficient to be precisely evaluated differently from other neoplastic conditions in childhood [59]. The SIOPEL Group recommends presurgical chemotherapy followed by the tumoral excision [60,61]. The liver transplantation should be reserved either initially for the non-excisable tumor or after the relapse. Tumor response to chemotherapy relies, at least partially, on histopathology, specifically on the relationship between the extent of immature and mature tumor elements. Patients with an excellent response to chemotherapeutic drugs have a better survival value despite a more advanced PRETEXT stage than children with HB with a lower PRETEXT stage and poor response to chemotherapeutic drugs [62]. However, children with PRETEXT IV HB and no extrahepatic disease should probably receive a total hepatectomy and liver transplantation. Such an approach aims for a prognosis rate assessed to be 70–80% [63]. 

## 2. Conclusions and Future Directions

Liver neoplasms remain a rare tumor category in childhood, but the exact recognition has deep and noteworthy prognostic implications. The ‘teratoid’ HB remains a rare neoplasm, which requires further investigation, probably comparing similar studies in ‘teratoid’ Wilms’ tumor and other pediatric neoplasms with ‘teratoid’ features. Recently, several investigations showed the efficacy of active blocking of the pathway of programmed death 1 (PD-1)/programmed death-ligand 1 (PD-L1) of immune checkpoint inhibitors in patients with various PD-L1-expressing cancers. In PD-L1 expressing neoplasms, anti-PD-1 antibodies can neutralize the inhibitory path that prevents effective antitumor T-cell responses. The result is to induce an antitumor effect. Still, most pediatric tumors have low expression of these proteins. In two HB only, intense PD-L1 staining was observed [64]. Thus, it should be studied from case to case. Finally, we emphasize that international collaborative studies are tremendously needed, and a central histopathological review should be mandatory. The “Pediatric Liver Unresectable Tumor Observatory” module was established as a joint register office supported by SIOPEL (Societe Internationale d’Oncology Pediatrique), IPTA (International Pediatric Transplant Association), COG (Children’s Oncology Group), GPOG (German Pediatric Oncology Group), and other groups to help reply some of the compelling questions that can arise in the nearest future. (http://pluto.cineca.org/, accessed on 31 March 2022).

## Figures and Tables

**Figure 1 children-09-00565-f001:**
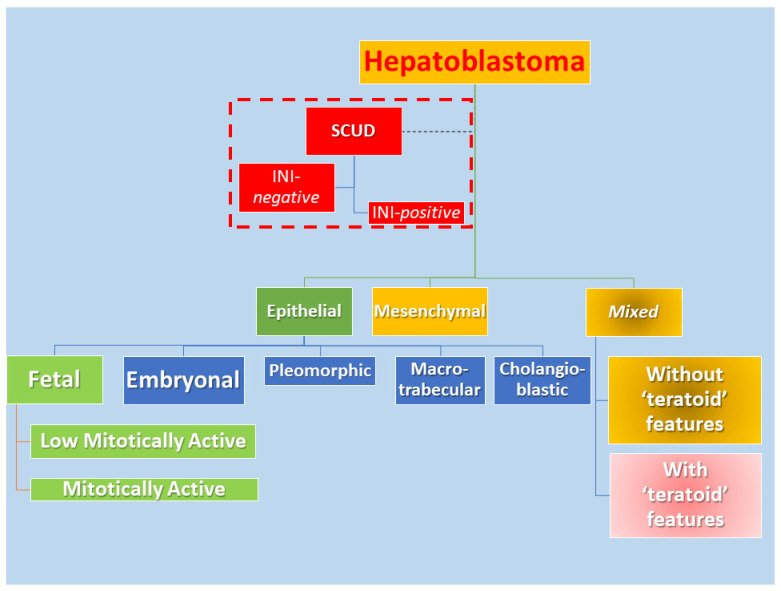
The schema depicts the most current classification of the subgroups of hepatoblastoma. The teratoid hepatoblastoma or hepatoblastoma with teratoid features is part of the mixed type (epithelial and mesenchymal components) of hepatoblastoma.

**Figure 2 children-09-00565-f002:**
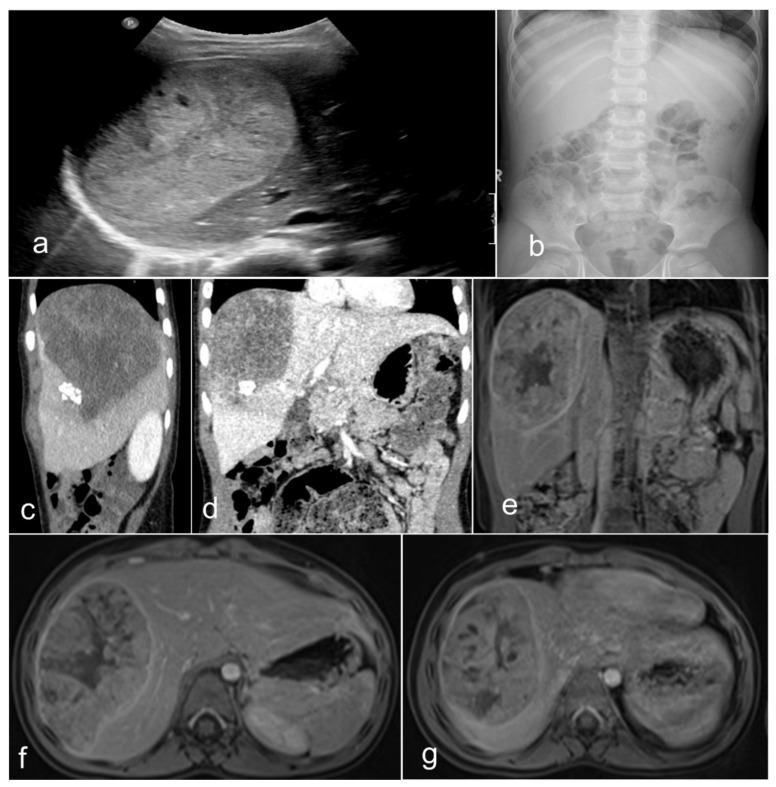
Imaging of a teratoid hepatoblastoma showing a heterogeneous mass encased in the liver (**a**), which is very close to the diaphragm (**b**). The magnetic resonance imaging (**c**–**g**) shows a large neoplasm in the right upper quadrant of the abdomen deriving from the right liver lobe. The neoplasm has two main portions and seems to contain different portions of tissue. The coronal view showed that the tumor is exophytic, protruding superiorly (T2 sequence). Remarkably, the transversal view typically reveals that the ventral tumor portion has a cystic nonenhancing appearance, whereas the dorsal part emerges more solid (T2 sequence). The T1 sequences—after contrast (post-gadolinium) application—(**e**–**g**) display an inhomogeneous contrast enhancement. The ventral portion of the hepatic neoplasm is intriguingly avidly enhancing, while the dorsal solid portion enhances less avidly.

**Figure 3 children-09-00565-f003:**
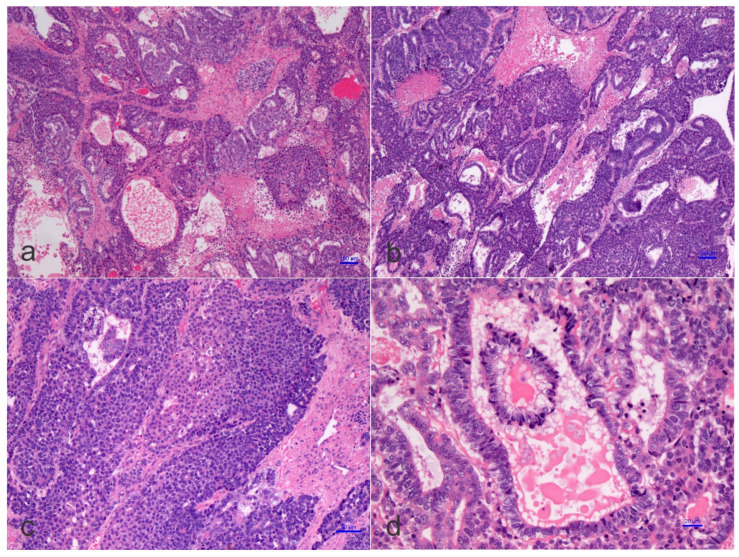
Histology of a mixed hepatoblastoma with teratoid features or teratoid hepatoblastoma showing fetal and embryonal portions and teratoid features with a yolk-sac-like pattern. Cords and solid portions are present and numerous mitoses are recognized with epithelial and mesenchymal components and yolk-sac-like elements (**d**), ((**a**–**d**) hematoxylin and eosin staining, X50 original magnification, scale bar, 100 microns).

**Figure 4 children-09-00565-f004:**
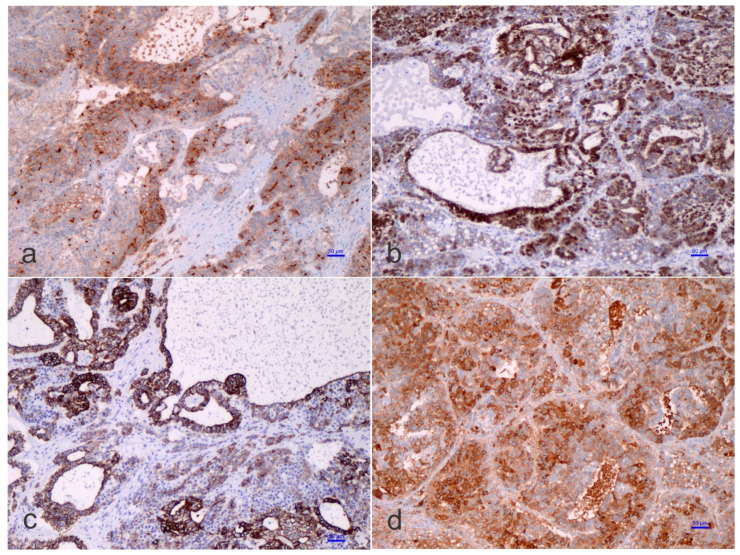
Immunohistochemistry of a teratoid hepatoblastoma. The microphotographs show positivity for polyclonal carcino-embryonic antigen (pCEA) ((**a**), Avidin-Biotin Complex anti-pCEA immunostaining, ×100 original magnification, scale bar, 50 microns), β-catenin ((**b**), Avidin-Biotin Complex anti-β-catenin immunostaining, ×100 original magnification, scale bar, 50 microns), (cyto-)keratin 19 (CK19) ((**c**), Avidin-Biotin Complex anti-K19 immunostaining, ×100 original magnification, scale bar, 50 microns), and α-fetoprotein (AFP) ((**d**), Avidin-Biotin Complex anti-AFP immunostaining, ×100 original magnification, scale bar, 50 microns).

**Figure 5 children-09-00565-f005:**
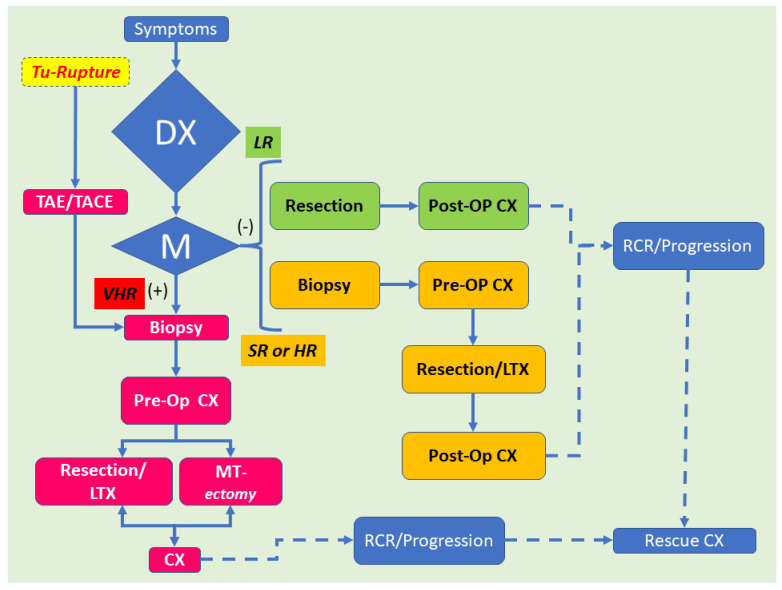
Diagnosis and therapeutic algorithm for hepatoblastoma. Hepatoblastoma is typically diagnosed by objective clinical signs, imaging data, and an increase in serum α-fetoprotein (AFP) levels. In some children affected with hepatoblastoma harboring a tumor, which is identified as ruptured, transarterial embolization (TAE) or transarterial chemoembolization (TACE) is performed to monitor the intraperitoneal hemorrhage. After the hemorrhage is under control, these children should be handled according to the very high-risk stratification flow. Amongst the children without distant tumor spreading [M(−)], low-risk (LR) children are approached with primary resection followed by chemotherapy in the postoperative period (Post-op CX). Standard-risk (SR) (also known as “intermediate-risk” (IR)) or high-risk (HR) children with HB receive chemotherapy in the preoperative period (Pre-op CX). Then, these patients undergo primary tumor resection by hepatectomy or liver transplantation (LTX). The very high-risk (VHR) patients affected by HB and distant tumor spreading (M(+))receive Pre-op CX. Then, a patient whose distant tumor spreading is reduced by CX goes through primary tumor resection by hepatectomy. Alternatively, these patients may undergo LTX, followed by Post-op CX. However, a patient whose distant tumor spreading remains needs to go through “metastasectomy” or hepatectomy. LTX is typically generally for the patient devoid of distant metastasis. On the other side, LTX may be indicated when distant tumor spreading has obviously vanished. Finally, patients with recurrence (RCR) or neoplastic progression should undertake some kind of rescue chemotherapy, also known as salvage CX, which is a form of therapy given after a disease does not react to standard therapy.

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
