# Peer review of "‘Teratoid’ Hepatoblastoma: An Intriguing Variant of Mixed Epithelial-Mesenchymal Hepatoblastoma"

_children, 2022, doi:10.3390/children9040565_

Round 1
Reviewer 1 Report
In congratulate you for an interesting and coherent review of a histologic subtype of Hepatoblastoma. The article is well written and intestesting to read for scientists and clinicians in the field of pediatric oncology and surgery.
Author Response
Thank you. I am uploading our response to reviewers. Please the attachment.

Reviewer 2 Report
The authors present a review of teratoid hepatoblastoma. While the review covers the perspectives from a clinical aspect and a lot of the discussion is pertaining to hepatoblastomas in general, the authors do not throw light on any novel aspects of the disease or its biology. A lot of it is conjecture and does not dive deeply into the concepts of progenitor stem cells and their role in hepatoblastomas. There are some specific issues that need to be rectified:
- The authors claim that mucinous and squamous epithelium is part of teratoid histology when in actual practice, they are still considered as components of epithelial HB. The true criteria for teratoid component is neuroepithelium, melanin or more recently yolk sac like component and neuroendocrine components. Osteoid, cartilage, muscle are all parts of mesenchymal component but fibrous stroma is not as mentioned in the abstract and text.
- It is wrong to suggest that teratoid component or mesenchymal components are seen mainly after chemotherapy. Most of these are diagnosed in the US on biopsies. It is true that they appear more prominent after chemotherapy due to shrinkage and response of epithelial elements but they are not an effect of chemotherapy but rather non-responsive component of mixed HB.
- It is wrong to say that "teratoid component becomes more florid after chemotherapy" (line 80,81,82). The response to chemotherapy in these neoplasms is variable with some tumors showing complete resolution and some persistence post-therapy. Having said that, it is the epithelial component that determines the aggressiveness of the HB with the teratoid component only rarely being seen in metastases.
- Line 133,134 - stem cells are multipotential and less-differentiated by definition so it is redundant to apply those adjectives.
- Cholangioblastic hepatoblastoma is a variant of epithelial hepatoblastoma and not part of the teratoid concept. They should not be mistaken as such. Most of these arise from primitive "blastemal" cells around epithelial elements, a concept that is still preliminary.
- Small cell undifferentiated is no longer shown to be of significance in prognosis of HB and is not considered an important prognostic indicator based on experience from the COG. Most of the pure SCUD tumors in literature are malignant rhabdoid tumors.
- The concept of oval cells have traditionally applied to mouse models of liver cell regeneration. The equivalent of the oval cell in humans is not known (the oval cell itself is not thought to be present in human livers). These stem cells are thought to be bone marrow derived though the putative liver stem cell is still not identified.
- Line 179: "In his acumen" does not seem appropriate, maybe in his review or insight? Incidentally, the cells that Zimmermann was referring to most likely correspond to the so-called blastemal cells in HB and are capable of bidirectional differentiation.
- The distinction of teratoid HB from a teratoma is clear and needs the presence of an epithelial HB component to make that diagnosis. True teratomas and malignant germ cell tumors of the liver exist.
- It is difficult to prognosticate mixed hepatoblastomas, as in most cases the prognosis is determined by the epithelial components and more often an embryonal component or pleomorphic epithelial component. While mesenchymal components may be seen in metastatic tumors they are usually small component and intermixed with the epithelial component.
Author Response

(The authors gave the same response as above.)

Reviewer 3 Report
In the article Consolato Sergi et al provide an excellent review of Teratoid hepatoblastoma - a subtype of Mixed Epithelial-Mesenchymal Hepatoblastoma. The manuscript is consice, well written with all needed details and could serve as a compendium for pediatric oncologists and those involved in the treatment of childhood liver tumors. Hepatoblastoma in children is a rare disease. Collecting diagnostic data (including pathology and genetics), conducting clinical trials in small populations is difficult and takes time. That is why every published paper including reviews, clinical cases should be supported to stimulate further research and development.
I have no critical comments and recommend to publish it.
A small correction -fig 9, spelling error - Tu rupture - e is missing.
Author Response

(The authors gave the same response as above.)

Round 2
Reviewer 2 Report
Thank you for incorporating the changes